# Contextual Kernels for Task-Aware Fine-Tuning in Vision-Language Models

## Abstract

Vision-Language Models (VLMs) demonstrate impressive generalization capabilities due to their training on extensive datasets such as ImageNet. However, their performance can decline when faced with unfamiliar tasks. While downstream fine-tuning enhances adaptability, it often compromises inherent generalizability. To address this challenge, we propose a novel method that leverages contextual generation to improve task and class representation within a semantic space. Our approach utilizes VLMs to generate detailed contextual descriptions and develop Contextual Kernels (CK) for each class in the semantic space. Our method preserves the core features of VLMs by freezing fundamental components while extending a linear network for semantic kernel density projection. This approach significantly enhances model adaptability for real-world tasks. Despite robust zero-shot capabilities, we investigate the incorporation of additional training samples to further improve adaptability in dynamic Task Incremental Learning (TIL) scenarios. Each task's unique CK distribution acts as a fingerprint, facilitating high-performance TIL with minimal forgetting. We validate the efficacy of our framework through experiments on four TIL datasets, achieving state-of-the-art performance. Our findings indicate that the semantic space within the text mode encapsulates both the generalizability and adaptability of VLMs, thus paving the way for robust applications across diverse and evolving task environments. This work systematically balances generalizability and adaptability in VLMs, addressing a critical gap in current research.

## 1 Introduction

Vision-Language Models (VLMs) have propelled advancements in computer vision by enabling impressive zero-shot task capabilities. However, their adaptability to dynamic environments is constrained due to their initial design focus on specific tasks. While fine-tuning VLMs on downstream tasks enhances adaptability, it often compromises their generalizability. Continual learning addresses this by allowing VLMs to integrate new data while preserving previously acquired knowledge, enabling adaptation to new tasks without forgetting old ones. Within continual learning, Task Incremental Learning (TIL) is pivotal, as it handles a sequence of tasks with disjoint classes, contrasting with traditional supervised learning that assumes a static data distribution.

In TIL, the evolving data distribution can cause VLMs to forget previously learned classes when fine-tuning on new tasks due to a shift in focus. Recent trends in TIL leverage pre-trained VLMs to utilize robust feature representations within their extensive semantic space, achieving strong zero-shot performance across various multi-modal applications. Balancing generalizability and adaptability remains a challenge in machine learning. Performance decline during fine-tuning on downstream tasks can be attributed to semantic collapse—a phenomenon noted in domain generalization tasks Cho et al. (2023). While VLMs are trained for broad semantic spaces, downstream tasks often require narrowed semantic contexts for optimal performance. To illustrate, consider the semantic differences between the CALTECH and LABELME subsets from the VLCS dataset Torralba & Efros (2011). Our analysis highlights variations in style, viewpoint, and background context. For instance, CALTECH images generally have clear backgrounds, whereas LABELME samples exhibit complex backgrounds (Fig. 1). This suggests that when certain aspects are irrelevant to current tasks, VLMs must focus within the relevant semantic space. A promising approach in TIL is Learning to Prompt (L2P) Wang et al. (2021), which develops prompts to guide VLMs in new tasks. Despite its simplicity, L2P has

Figure 1: Impact of semantic distribution drift on TIL. The figure presents the CALTECH subset of the VLCS dataset. Significant differences between different domains in the semantic space, contrasting with the raw feature space. This observation underpins our approach to modeling task representation within the semantic distribution using kernel density-based feature projection.

achieved notable performance without rehearsal buffers. However, prompts in higher-dimensional spaces often lack explainability and face challenges in managing multiple tasks within TIL.

We hypothesize that each task's classes can be mapped into a shared semantic distribution space, where each task occupies a unique subspace defined by specific semantic contexts. We propose a novel Contextual Kernel Density-Based Task Representation Learning Framework that fine-tunes VLMs at test time using rich contextual information from test set samples. Our approach enables effective comparisons between tasks trained at different stages, even without overlapping training samples. Existing model generalization methods like PromptStyler Cho et al. (2023) and Mao et al. Mao et al. (2024) leverage zero-shot classification but overlook how additional training samples can enhance adaptability. Our method addresses this by filtering irrelevant samples during fine-tuning using CK distribution thresholds derived from the text modality. By excluding distracting contexts and emphasizing relevant ones, we significantly enhance model adaptability. Furthermore, our fine-tuned VLMs generate CK-based confidence scores during testing, allowing them to abstain from decisions on test samples outside predefined categories—a critical feature for safety-critical applications such as medical diagnostics and autonomous driving.

**Our contributions are as follows:**

- *Task and Class Representation Learning for TIL*: We introduce a framework that fine-tunes Vision-Language Models (VLMs) through context-based kernel density feature representation learning. This approach facilitates effective comparisons between non-overlapping training tasks and classes within the current task, leveraging distribution measures.

- *Mitigating Semantic Collapse*: We tackle the issue of semantic collapse by filtering out irrelevant contexts, thereby optimizing performance within narrower, task-specific semantic spaces. Each task demonstrates independent feature distribution patterns, enabling not only the classification of classes within a task but also the differentiation between non-overlapping tasks.

- *Enhanced Adaptability*: By leveraging language as a robust representation space, we enhance the generalizability and adaptability of VLMs. We generate confidence scores based on context knowledge (CK) that empower models to abstain from making decisions on non-categorical test samples, ensuring reliability in safety-critical applications.

## 2 BACKGROUND

This paper addresses the challenges of Task Incremental Learning (TIL), which focuses on developing models that sequentially learn tasks while minimizing catastrophic forgetting. TIL methods are categorized into regularization-based, rehearsal-based, and architecture-based approaches, with emerging prompt-based techniques leveraging Vision-Language Models (VLMs) to enhance parameter efficiency. Our contribution lies in proposing an end-to-end framework grounded in CK representation learning, optimizing task and class representations to improve performance and task separation. Additionally, we explore Kernel Density Function Based Representation Learning (KDF-RL), which projects data into high-dimensional spaces using kernel functions to capture complex relationships, making it particularly useful for tasks like anomaly detection and metric learning with probabilistic labels. Furthermore, we highlight the significance of semantic guidance in fine-tuning VLMs, especially in open set and zero-shot learning, where models utilize language embeddings for generalization to unseen classes. By integrating kernel-based techniques with these advancements, our work enhances representation learning for CK tasks, effectively managing uncertainties and improving model robustness across various learning scenarios. This comprehensive approach demonstrates significant potential for advancing TIL and related areas in machine learning. More detailed backgrounds can be found in Appendix A.9.

## 3 METHODS

### PROBLEM DEFINITION

Consider a sequence of tasks $D = \{D_1, D_2, \ldots, D_T\}$, where $T$ represents the total number of incremental tasks. Each $t^{th}$ task, denoted as $D_t = \{(\mathbf{x}_l^t, y_l^t)\}$, consists of data samples $\mathbf{x}_l^t \in X$ and their corresponding ground-truth labels $y_l^t \in Y$. The objective of continual learning is to develop a single model $f_\theta : X \to Y$, parameterized by $\theta$, that can effectively handle all $T$ incremental tasks.

During inference, the model $f_\theta$ must predict the label $y \in Y$ for a given sample $\mathbf{x}$, which may be unseen from any of the tasks. A significant challenge arises when the task identifier is absent for a test instance. In such scenarios, the prediction probability can be decomposed into two distinct probabilities: the *Within-Task Prediction* (WP) and the *Task Prediction* (TP). This relationship is formulated as follows:

$$\mathbf{P}(i|\mathbf{x}) = \mathbf{P}(i|\mathbf{x}, t) \cdot \mathbf{P}(t|\mathbf{x}), \tag{1}$$

where $i$ denotes the class label and $t$ represents the task identifier. The first term on the right-hand side corresponds to WP, while the second term represents TP.

In this paper, we aim to design a model $f_\theta$ that demonstrates both generalizability across all tasks and adaptability to specific contexts within each task. For clarity and consistency, we will refer to Appendix A.4 for each symbol according to its corresponding meanings.

### 3.1 THE TASK REPRESENTATION LEARNING FRAMEWORK

In this work, we introduce a framework for task and class representation learning within a semantic space, leveraging kernel density estimation in text embeddings to harness rich contextual information. As illustrated in Fig. 2, we consider a task with three classes centered around the theme of art style. The task representation is designed to model the mixture of class distributions, serving as a unique fingerprint that differentiates it from other tasks. The anchors for this representation are derived from text modal class distributions, utilizing detailed language descriptions provided by large multimodal models (LMMs). Our framework employs a projection network that learns to effectively separate different classes by establishing a defined probability margin. This network brings training images of the same class closer to their corresponding text modal distributions while characterizing the task representation through the mixture of class probability density functions (PDFs).

Task-specific contexts are generated using vision-language models (VLMs), leveraging their inherent generalizability. We have developed an extensible context prompt pool that encompasses 11 categories, detailed in Appendix A.1, each containing a comprehensive list of fine-grained contexts. By utilizing these task-specific contexts as semantic guidance, we sample points within the text embedding space

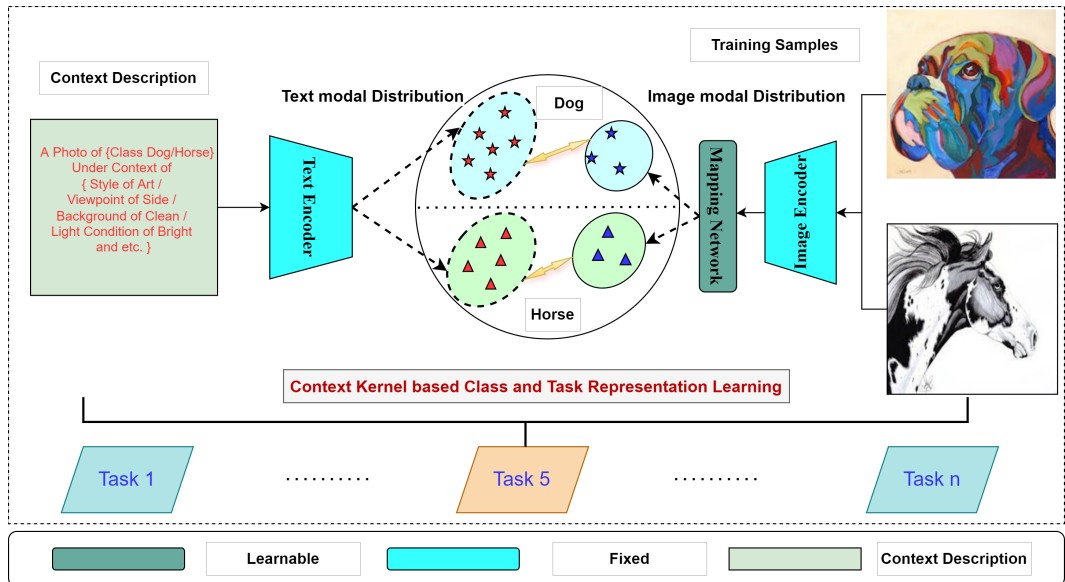

Figure 2: Overview of the context kernel PDF based task representation learning in CIL. The above graph depicts a task under a specific context of style and background, where there are two classes. The task representation aims to model the mixture of class distribution as a fingerprint for the task to differentiate from other tasks. The text modal distribution for each class is derived by utilizing detailed language descriptions within LMMs, which are represented in the figure as red markers: stars and triangles. The image modal distribution is denoted in blue color. We employ our task representation learning to train a mapping network, which effectively separates different classes by a defined margin while bringing distributions in text and image modal of the same class closer.

for each class. During the training phase, the projection network is tasked with learning a semantic representation that pulls image distributions toward their corresponding text modal anchors while pushing apart the distributions of different classes. The primary objective is to cluster samples of the same class closely together in the semantic space, ensuring a substantial margin separates samples from different classes. For instance, as depicted in Fig. 2, contextualizing classes such as dogs and horses under the domain of art allows them to occupy distinct regions within the semantic space. This separation is crucial for enhancing the model's adaptability, mitigating the impact of extraneous contextual disturbances.

To refine the task and class representations, the projection network aligns training samples in the image modality with their corresponding class distributions in the text modality, while concurrently maintaining significant separation between different classes. This approach ensures that our model not only generalizes well across various tasks but also adapts effectively to specific contexts within each task, ultimately enhancing overall performance.

## 3.2 CLASS REPRESENTATION IN SEMANTIC SPACE

The TIL addresses the challenge of evolving feature representations due to changing data distributions across different tasks. While the image features may vary, the task or class semantics in the text model often remain stable. Language has evolved over thousands of years to provide a compact semantic representation of the world, allowing models to leverage semantic cues, such as class names and contextual information, from both current and previous tasks without additional cost. To facilitate task comparison and class prediction, we propose transforming all feature representations from a multimodal setup into a unified semantic text embedding space. Specifically, we utilize a text encoder of VLMs to encode the task-related context and class knowledge into a unified CK space.

For a given task $t$, we denote the class names relevant to this task by the set $\mathcal{Y}^t$. The objective of task $t$ is to accurately classify the classes represented in $\mathcal{Y}^t$. We represent the language context of the task through prompts composed of class names, structured as follows:

"A photo of {**class i**} in {**context j**} $\cdots$"

In this format, the placeholder {**class i**} is replaced with the corresponding class names for the task. During testing, class labels are not utilized; instead, only the contextual information is employed to derive the CK. The constructed prompts serve as inputs to the VLMs, enabling the extraction of embedding text features corresponding to the output tokens. This results in a text representation $D_t \in \mathbb{R}^{N_t \times d}$, where $d$ represents the embedding dimension and $N_t$ denotes the number of generated samples for task $t$.

Let $\mathbf{x}_{ijl}$ denote the embedding vector corresponding to the $l^{\text{th}}$ test image within the $i^{\text{th}}$ class and the $j^{\text{th}}$ context, where $i \in \{1, 2, \ldots, C\}$ and $C$ represents the total number of classes. Additionally, let $N_j$ denote the size of the context pool, while $N_i$ indicates the number of test samples used to determine the task-specific contexts. By leveraging the rich contextual information encapsulated in the sampled points, we aim to enhance the precision of the CKs. This approach ultimately contributes to improved performance in TIL scenarios. It facilitates a more refined understanding of the underlying task semantics and allows for effective adaptation of the model to new tasks without compromising performance on previously learned ones.

The mean vector $\boldsymbol{\mu}_i$ and the variance $\boldsymbol{\sigma}_i^2$ for each class $i$ are calculated as follows:

$$\boldsymbol{\mu}_i = \frac{1}{N_j * N_i} \sum_{j=1}^{N_j} \sum_{l=1}^{N_i} \mathbb{1}_{\{\text{context}=j\}} \mathbf{x}_{ijl} \tag{2}$$

$$\boldsymbol{\sigma}_i^2 = \frac{1}{N_i * N_j - 1} \sum_{j=1}^{N_j} \sum_{l=1}^{N_i} \mathbb{1}_{\{\text{context}=j\}} (\mathbf{z}_{ijl} - \boldsymbol{\mu}_i)^2 + \epsilon \tag{3}$$

where $\mathbf{x}_{ijl}$ denotes the embedding feature for the $l^{th}$ test image in class $i^{th}$ from context $j^{th}$ based on the text embeddings. The indicator function $\mathbb{1}_{\{\text{context}=j\}}$ signifies the activated values for the $j^{\text{th}}$ context within an instance or batch-level test set, derived from VLMs. The term $N_i \times N_j$ represents the product of the number of test samples $N_i$ and the number of predefined contexts $N_j$. The parameter $\epsilon$ denotes the uncertainty that we introduce into the variance to enhance the model's generalization capabilities.

Our methodology systematically generates and leverages task-specific contexts derived from VLMs, represents CKs within the semantic embedding space, and utilizes various categories of contexts to enhance the representation and understanding of each class within a given task. This approach ensures robust and precise context determination, which is critical for advanced visual scene understanding and nuanced content analysis.

### 3.3 THE DISTRIBUTION LOSS

To express the distribution for the classes in a task, We use the kernel function as follows to evaluate the probability of a training sample $x_s$ in the image modality with respect to $x_{text}$ in the text distribution $D_i$:

$$\mathbf{K}(\mathbf{x}_s) = \frac{1}{N_i * h^d} \sum_{x_{text} \in D_i} \mathbf{K}\left(\frac{\mathbf{x}_s - \mathbf{x}_{text}}{h}\right) \tag{4}$$

The bandwidth $h$ is a hyper-parameter applied to each dimension. In Eq. 4, $\mathbf{x}_s$ denotes a training sample in the image modal for which the CK is computed against the text modality embeddings. Conversely, $\mathbf{x}_{text}$ represents the anchor points $D_i$ in the text modality, drawn from rich contexts for class $i$. For each class associated with a given task, we sample $N_i$ text embeddings to serve as these anchor points. In this section, we propose our learning objective to act as the training loss, replacing the conventional cross-entropy loss and guiding the network training process.

$$\mathcal{L}(\mathbf{L}) = \max(- \sum_{x_{text} \in D_i} \mathbb{1}_{\{y=i\}} \mathbf{K}(\mathbf{x}_s - \mathbf{x}_{text})$$
$$+ \sum_{x_{text} \in D, x_{text} \notin D_i} \mathbb{1}_{\{y \neq i\}} (\mathbf{K}(\mathbf{x}_s - \mathbf{x}_{text}) + \Delta, 0) \tag{5}$$

In Eq. 9, the set of trainable parameters, denoted by $\mathbf{L}$, is implemented as a linear projection network. Here, $\mathbf{x}_s$ denotes the features of a training sample in the image modality, while $\mathbf{x}_{text}$ signifies the anchor CK vectors within the $y^{th}$ class in the semantic embedding. The symbol $\Delta$ represents the CK margin, ensuring that the CK for positive samples exceeds that of negative instances by a safe margin. Eq. 9 is utilized as the loss function in our framework. The probability values involved in the loss computation for each anchor are expressed in logarithmic format, which stabilizes the training process and prevents underflow during backpropagation. More kernal based metric learning backbround can be found in Appendix A.10.

### 3.4 TASK PREDICTION AND WITHIN TASK CLASS PREDICTION

During the testing stage, the task label for each test instance is determined using the *TP* procedure, where only the image embedding features are utilized. The prediction process is formalized as follows:

$$\mathcal{T} = \arg\max_t \sum_{i \in \mathcal{Y}^t} \arg\max_i \mathbf{K}_i(\mathbf{x}_s), \tag{6}$$

where $t$ represents the index of the previously trained task. The index $i$ denotes the class label within task $t$, while $\mathcal{T}$ indicates the predicted task ID. The function $\mathbf{K}_i(\mathbf{x}_s)$ provides the semantic projection for the test sample $\mathbf{x}_s$ in class $i$. The set $\mathcal{Y}^t$ comprises the non-overlapping classes associated with task $t$. The CKs for each class, along with the CKs for each task, serve as distinctive fingerprints that characterize the respective task and class identities within the semantic space. This representation aids in accurately determining the task and class labels during the testing phase.

For *WP*, to assign an observation $\mathbf{x}_s$ to each of the classes of $\mathbf{Y}$ can be solved by maximizing the conditional probability given task label $\mathcal{T}$:

$$P[Y = i | \mathbf{x}_s, \mathcal{T}] = \frac{\mathbf{K}_i(\mathbf{x}_s)}{\sum_{i' \in \mathcal{Y}^{\mathcal{T}}} \mathbf{K}_{i'}(\mathbf{x}_s)}, \tag{7}$$

where $i' \in \mathcal{Y}^{\mathcal{T}}$ is the all classes in task $\mathcal{T}$.

According to Eq. 10, we can obtain the *TP* and select the appropriate model corresponding to the task $\mathcal{T}$. Subsequently, we utilize Eq. 11 to derive the *WP* for obtaining the class label. Thus, the procedure defined in Eq. 1 for a test instance is implemented within a TIL framework.

Since our prediction is represented as a probability, it is straightforward to establish a threshold to filter out samples that experience significant semantic shifts. During the testing stage, we also apply a threshold value determined by the lowest semantic CK value in the text modality for the corresponding class. If the mapped semantic CK value falls below this threshold, the image will not be predicted, indicating that it does not belong to any of the predefined categories within the specific context of the test scenario.

## 4 EXPERIMENTS

### DATASETS, EVALUATION METRICS AND EXPERIMENTAL SETTINGS

The CIFAR-100 dataset, ImageNet-Rendition (ImageNet-R), TinyImageNet, and ImageNet100 are used to evaluate TIL performance. These datasets offer diverse and challenging visual contexts for assessing model adaptability and robustness. Average Accuracy and Forgetting are common metrics employed to measure TIL performance, with higher average accuracy indicating better

| Method | $B$ | 5 Tasks | | 10 Tasks | | 20 Tasks | |
|---|---|---|---|---|---|---|---|
| | | $A_a \uparrow$ | $F \downarrow$ | $A_a \uparrow$ | $F \downarrow$ | $A_a \uparrow$ | $F \downarrow$ |
| DER++ Buzzega et al. (2020) | 1000 | - | - | 55.47 | 34.64 | - | - |
| BiC Wu et al. (2019) | 1000 | - | - | 52.14 | 36.7 | - | - |
| ER Chaudhry et al. (2019a) | 1000 | - | - | 55.13 | 35.38 | - | - |
| $Co^2L$ Cha et al. (2021) | 1000 | - | - | 53.45 | 37.3 | - | - |
| EWC Kirkpatrick et al. (2017) | 0 | - | - | 35.00 | 56.16 | - | - |
| LwF Li & Hoiem (2017) | 0 | 40.62 | 50.69 | 38.54 | 52.37 | 32.05 | 53.42 |
| L2P Wang et al. (2022c) | 0 | 62.61 | 8.01 | 61.21 | 8.65 | 57.36 | 9.07 |
| DualPrompt Wang et al. (2022b) | 0 | 67.83 | 4.79 | 66.47 | 5.75 | 63.25 | 6.13 |
| Coda-P Smith et al. (2023) | 0 | 75.25 | 6.86 | 74.26 | 7.91 | 71.16 | 8.49 |
| PC Dai et al. (2024) | 0 | 75.41 | 6.42 | 74.34 | 7.35 | 71.44 | 7.62 |
| Ours (CK) | 0 | **78.85** | **4.55** | **78.20** | **5.65** | **77.65** | **6.10** |
| Upper-bound | 0 | 79.31 | - | 79.31 | - | 79.31 | - |

Table 1: Performance comparison on the ImageNet-R dataset for TIL. $B$ denotes buffer size. Prompt-based methods use an instance-wise setup.

performance and lower forgetting indicating better retention of previously learned knowledge. The evaluation follows established benchmarks and experimental settings to ensure meaningful comparisons with existing literature. This approach enables a robust assessment of the proposed TIL methods' performance and robustness. The detailed description can be found in Appendix A.2.

### 4.1 PERFORMANCE EVALUATION

In this study, we conduct a comprehensive comparison of various well-established approaches to TIL. These approaches include regularization-based methods such as EWC Kirkpatrick et al. (2017) and LwF Li & Hoiem (2017); rehearsal-based techniques including ER Chaudhry et al. (2019a), BiC Wu et al. (2019), DER++ Buzzega et al. (2020), and $Co^2L$ Cha et al. (2021); and prompt-based strategies like L2P Wang et al. (2022c), S-Prompt Wang et al. (2022a), DualPrompt Wang et al. (2022b), CODA Smith et al. (2023), ESN Wang et al. (2023), DAP Jung et al. (2023), PC Dai et al. (2024), and our proposed method denoted as CK. To ensure fair comparisons, all methods utilize a pre-trained ViT-B/16 as the backbone and adhere to the settings established in Dai et al. (2024). The upper-bound performance is derived from supervised fine-tuning on i.i.d. data from all tasks, representing the best achievable benchmark for any continual learning method. We use Average Accuracy ($A_a$) and Forgetting ($F$) as performance metrics.

Tables 1 and Tables 2 illustrate that our proposed method consistently outperforms existing techniques, particularly as the number of tasks increases. The tables provided showcase the performance of various methods on the ImageNet-R and CIFAR-100 datasets in a TIL setting, focusing on accuracy (denoted as $A_a$) and forget rate (denoted as $F$) across different task increments (5, 10, and 20 tasks) while varying buffer sizes. Overall, it is evident that traditional methods like DER++, BiC, and ER generally show lower accuracy and higher forget rates compared to more recent approaches. As the number of tasks increases, many methods exhibit a drop in accuracy and an increase in forget rates, indicating the common challenge of catastrophic forgetting in continual learning scenarios. In contrast, advanced techniques such as DualPrompt, Coda-P, and PC demonstrate notable improvements, particularly in reducing forget rates while maintaining competitive accuracy. However, even these methods fall short compared to our proposed method, CK, which consistently achieves the highest accuracy across all task settings in both datasets. For instance, CK attains an accuracy of 78.85 for 5 tasks on CIFAR-100, outperforming the nearest competitor, PC, by a significant margin, while also excelling in forget rate with a score of 4.55 for 5 tasks. This indicates that CK effectively retains learned information from previous tasks while integrating new tasks, a crucial aspect of continual learning. Furthermore, as the number of tasks increases, CK maintains superior performance, with accuracy remaining above 77% even at 20 tasks, showcasing its robustness against the challenges posed by incremental learning settings. In summary, CK not only leads in performance metrics but also addresses one of the significant challenges in continual learning—catastrophic forgetting—setting a new benchmark for future research in class-incremental learning.

| Method | $B_s$ | 5 Tasks | | 10 Tasks | | 20 Tasks | |
|---|---|---|---|---|---|---|---|
| | | $A_a \uparrow$ | $F \downarrow$ | $A_a \uparrow$ | $F \downarrow$ | $A_a \uparrow$ | $F \downarrow$ |
| DER++ Buzzega et al. (2020) | 1000 | - | - | 55.47 | 34.64 | - | - |
| BiC Wu et al. (2019) | 1000 | - | - | 52.14 | 36.7 | - | - |
| ER Chaudhry et al. (2019a) | 1000 | - | - | 55.13 | 35.38 | - | - |
| $Co^2L$ Cha et al. (2021) | 1000 | - | - | 53.45 | 37.3 | - | - |
| EWC Kirkpatrick et al. (2017) | 0 | - | - | 35.00 | 56.16 | - | - |
| LwF Li & Hoiem (2017) | 0 | 40.62 | 50.69 | 38.54 | 52.37 | 32.05 | 53.42 |
| L2P Wang et al. (2022c) | 0 | 62.61 | 8.01 | 61.21 | 8.65 | 57.36 | 9.07 |
| DualPrompt Wang et al. (2022b) | 0 | 67.83 | 4.79 | 66.47 | 5.75 | 63.25 | 6.13 |
| Coda-P Smith et al. (2023) | 0 | 75.25 | 6.86 | 74.26 | 7.91 | 71.16 | 8.49 |
| PC Dai et al. (2024) | 0 | 75.41 | 6.42 | 74.34 | 7.35 | 71.44 | 7.62 |
| Ours (CK) | 0 | **78.85** | **4.55** | **78.20** | **5.65** | **77.65** | **6.10** |
| Upper-bound | 0 | 79.31 | - | 79.31 | - | 79.31 | - |

Table 2: Performance comparison on the split CIFAR-100 dataset under the class-incremental learning setting. $B_s$ denotes the buffer size. Results marked with $\star$ are sourced from the original papers, $\dagger$ from Wang et al. (2022b), and $\ddagger$ are computed using the respective codebases and standard evaluation metrics. Prompt-based methods are evaluated in an instance-wise prompt setup.

Table 3: Mean Average Accuracy under large number tasks settings. The CIFAR100 is split into 20 and 50 tasks (C-20T and C-50T), the TinyImageNet is split into 50 and 100 tasks (T-50T and T-100T), and the ImageNet100 is split into 50 tasks (I-50T). The compared method is DyTox, as only this method reports performance under large number of task splits greater than 50.

| Method | C-20T | C-50T | T-50T | T-100T | I-50T |
|---|---|---|---|---|---|
| | Mean $\pm$ Std | Mean $\pm$ Std | Mean $\pm$ Std | Mean $\pm$ Std | Mean $\pm$ Std |
| DyTox | $72.27_{\pm 0.32}$ | $70.20_{\pm 1.97}$ | - | - | - |
| Ours (CK) | $\mathbf{86.65}_{\pm 0.45}$ | $\mathbf{85.20}_{\pm 0.45}$ | $\mathbf{83.44}_{\pm 0.50}$ | $\mathbf{81.25}_{\pm 0.50}$ | $\mathbf{84.6}_{\pm 0.25}$ |

# 5 DISCUSSION

## 5.1 PERFORMANCE ADVANTAGE ON NUMEROUS TASK SETTINGS

To evaluate the effectiveness of our method under conditions involving a large number of task splits, we conducted comprehensive experiments on the TinyImageNet, ImageNet100, and CIFAR100 datasets. These datasets, characterized by a high number of classes, are well-suited for configurations with 50 or more task splits. Given that only the DyTox method Douillard et al. (2022) has reported performance under such conditions, we focus our comparative analysis on this model. Our learning framework, CK, demonstrates a significant advantage in managing numerous tasks, as evidenced in Table 3. Our method maintains stable average accuracy levels even as the number of tasks increases to 100. This stability is crucial in the context of TIL, where performance maintenance amidst growing complexity is essential.

On the CIFAR100 dataset with 20 task splits (C100-20T), our method achieves an impressive mean accuracy of **86.65%**, significantly outperforming DyTox, which reports only 72.27%. As the number of task splits increases to 50 (C100-50T), our method continues to excel with a mean accuracy of **85.20%**, while DyTox's performance drops to 70.20%. This trend of superior performance is also observed in the TinyImageNet dataset, where CK achieves **83.44%** and **81.25%** mean accuracies for 50 and 100 task splits (T-50T and T-100T), respectively, underscoring its exceptional scalability. Moreover, for ImageNet100 with 50 task splits (I-50T), our method CK maintains a commendable mean accuracy of **84.6%**. The observed correlation between the number of task splits and accuracy levels further underscores the efficacy of our approach. While DyTox employs dynamic token expansion to address the inherent challenges in TIL, our method not only matches but significantly exceeds its performance, particularly in terms of Top-1 mean accuracy.

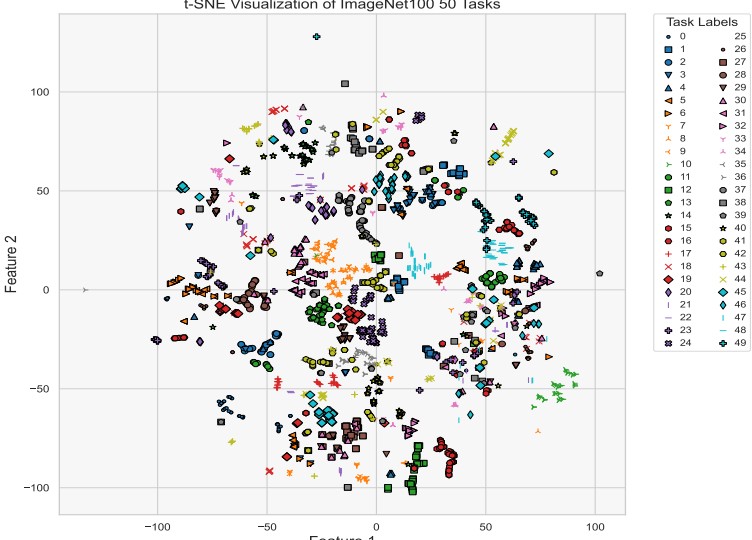

Figure 3: The t-SNE visualization of the ImageNet100 test samples in the CK space demonstrates clear separation of the 50 tasks. The embedding semantic features, projected into a 2D space, distinctly separate each task, with different markers denoting different tasks. The excellent task prediction performance in the semantic space helps the model achieve superior TIL accuracy.

## 5.2 TASK REPRESENTATION VISUALIZATION IN SEMANTIC SPACE

Traditional image feature embedding techniques often fall short in capturing the nuanced distinctions between different tasks. Conversely, in the semantic space, tasks are naturally situated in distinct semantic regions, a phenomenon likely rooted in the historical evolution of language. To substantiate this hypothesis, we present a 2D t-SNE (t-distributed Stochastic Neighbor Embedding) visualization, as depicted in Fig. 3. This visualization leverages test samples from the ImageNet100 dataset, with each sample labeled according to its corresponding task. Remarkably, the tasks are distinctly separated within the semantic space, with each task and its associated instances occupying unique regions in the graph. This clear demarcation underscores the effectiveness of our method in addressing the TIL challenge.

Furthermore, our model demonstrates a competitive edge over zero-shot models which fully leverage the generalizability of VLMs without necessitating additional training samples for model adaptation. Our approach also permits training on specific samples that closely resemble the test set, thereby enhancing adaptability. In scenarios where training samples are significantly different from the test instances within the semantic space, these samples can be effectively filtered out, ensuring that the model's intrinsic generalizability remains intact. The rich contextual information available at test time further enhances our model's adaptability, allowing it to excel in settings with a large number of tasks. This capability is crucial for effectively navigating the complexities and variances inherent in diverse task environments.

To validate the effectiveness of the within-task prediction among the classes, we visualized the semantic embedding space of the 100 classes in the ImageNet100 dataset. As depicted in Appendix A.3, all the classes in the dataset are distinctly positioned in the projected space and exhibit clear clustering properties. These well-separated clusters within the semantic embedding space facilitate the overall TIL process, enabling the model to achieve high accuracy in continual learning settings. The distinctiveness of the semantic positions and the strong clustering behavior observed in the visualization highlight the robustness of the embedding space. This separation ensures that the model can effectively distinguish between different classes during incremental learning, thereby improving its performance over multiple tasks. Consequently, the model's ability to maintain and generalize knowledge across tasks is significantly enhanced, leading to optimal TIL accuracy.

## 5.3 ABLATION STUDY

In our ablation study conducted on the CIFAR100 dataset with 20 task-split settings, we sought to evaluate the impact of pre-trained models, rich-context information, and the kernel density-based representation learning (KD-RL) method. The study involved two pre-trained models, ViT-B/16 and ResNet-50, both of which were pre-trained on a subset of ImageNet classes that deliberately excluded those overlapping with CIFAR and TinyImageNet. This selection aimed to leverage robust feature representations.

The role of rich-context information, extracted from the test set, was found to significantly enhance CIL, particularly when combined with either of the pre-trained models. Notably, when ViT-B/16 was utilized in conjunction with KD-RL, it achieved a superior performance of 88.65, underscoring its pivotal role. This result emphasizes the importance of a robust preliminary feature representation, which is crucial for KD-RL to effectively differentiate tasks.

Conversely, the KD-RL method was unable to demonstrate its potential when used with ResNet-50, resulting in inferior performance outcomes. This shortfall can be attributed to the inadequate feature representation capability of ResNet-50, which limited KD-RL's ability to construct task representations based on raw features from task-related classes. In conclusion, the combination of ViT-B/16 with rich-context and KD-RL demonstrates a marked improvement in CIL, highlighting the necessity of robust feature extraction for effective task differentiation. The limitations observed with ResNet-50 further underscore the critical nature of initial feature quality for the success of KD-RL.

Table 4: Ablation Study on CIFAR100 under 20 Task Splits

| ViT-L/16 | ViT-B/16 | ResNet-50 | Rich-Context | KD-RL | $A_a \uparrow$ |
|:---:|:---:|:---:|:---:|:---:|:---|
| ✓ | | | | | 74.50 |
| ✓ | | | ✓ | | 79.35 |
| ✓ | | | ✓ | ✓ | **91.65** |
| | ✓ | | | | 70.10 |
| | ✓ | | ✓ | | 76.20 |
| | ✓ | | ✓ | ✓ | 88.65 |
| | | ✓ | | | 65.50 |
| | | ✓ | ✓ | | 70.30 |
| | | ✓ | ✓ | ✓ | 50.15 |

## 6 CONCLUSION

This paper presents significant advancements in TIL through innovative semantic-based projection and kernel density based distribution learning methods. Our approach enhances model adaptability by fine-tuning within a narrowed semantic space, strategically focusing on relevant classes and contexts to mitigate semantic collapse and task confusion. The introduction of CKs leverages contextual information, refining the projection of classes into unique subspaces within a shared semantic distribution, which are class PDF anchors in text modal. This not only improves performance across various stages of TIL but also addresses the challenges of explainability and adaptability in high-dimensional semantic spaces without the need for rehearsal buffers. The robustness of our method is underscored by its ability to filter out irrelevant samples during fine-tuning, ensuring that the model retains focus on pertinent information. Additionally, the integration of confidence scores enables informed decision-making, allowing models to abstain from classifying out-of-category samples—an essential feature for safety-critical applications such as medical diagnostics and autonomous driving. Comprehensive experiments across four TIL settings validate the effectiveness of our approaches, achieving state-of-the-art results. These contributions pave the way for future research in adaptive learning systems, emphasizing the importance of contextual understanding and semantic clarity in dynamic environments. Our findings open new avenues for enhancing model performance through strategic projection methods and contextual awareness, encouraging further advancements in the field of continual learning.

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

## A  APPENDIX

### A.1  DETAILED CONTEXTS FOR EACH TYPE

The base contexts are summarized as:

- **Viewpoints:** Side View, Top View, Front View, Rear View, Three-Quarter View, Bottom View, Oblique View, Close-Up View, Distant View.

- **Styles:** Art, Painting, Sketch, Drawing, Picture (Photograph), Cartoon, Illustration, Diagram, Digital Art, Black and White, Colorized, Abstract Art, Realistic, Surrealistic, Impressionistic, Minimalistic, Vintage, Modern.

- **Backgrounds:** Natural Landscape, Urban Environment, Indoor Scene, Sky, Water Bodies (Sea, Lake, River), Forest, Mountain, Desert, Beach, Snow, Grassland, Field or Farmland, Park, Street, Building Interior, Office Space, Home Interior, Garden, Vehicle Interior, Sports Field or Arena, Commercial Space (e.g., shop, mall), Industrial Area, Rural Area, Underwater, Cave, Laboratory, School or Classroom, Hospital, Airport, Train Station, Construction Site, Amusement Park, Historical Site, Religious Building (e.g., church, mosque), Forest Path, Playground, Bridge, Camping Site, Parking Lot, Market or Bazaar.

- **Lighting Conditions:** Natural Light, Artificial Light, Daylight, Sunset, Sunrise, Nighttime, Dawn, Dusk, Overcast, Sunny, Partly Cloudy, Indoor Lighting, Fluorescent Light, Incandescent Light, LED Light, Candlelight, Street Light, Spotlight, Stage Light, Flash Photography, Low Light, High Contrast Lighting, Soft Lighting, Harsh Lighting, Backlighting, Front Lighting, Side Lighting, Diffused Light, Shadow Presence, Reflection Light, Ambient Light, Twilight.

- **Color Schemes:** Grayscale, Full Color, Monochrome, Sepia, High Saturation, Low Saturation, Black and White, Warm Colors, Cool Colors, Pastel Colors, Neon Colors, Muted Colors, Vibrant Colors, Duotone, Multicolor, Vintage Color, Pop Art Colors, Analogous Colors, Complementary Colors, Triadic Colors, Tetradic Colors, Split-Complementary Colors, Neutral Colors, Earth Tones, Rainbow Colors.

- **Environmental Conditions:** Indoor, Outdoor, Sunny, Cloudy, Rainy, Snowy, Windy, Foggy, Stormy, Hazy, Dusty, Humid, Dry, Hot, Cold, Misty, Icy, Clear Skies, Partly Cloudy, Thunderstorm, Blizzard, Sandstorm, Wet, Smoky, Frosty, Polluted, Calm, Breezy, Tornado, Hurricane.

- **Resolutions:** Low Resolution, Medium Resolution, High Resolution, Ultra-High Resolution, Thumbnail, Standard Definition (SD), High Definition (HD), Full HD (FHD), 4K Resolution (UHD), 8K Resolution, Blurred, Sharp, Pixelated, Compressed, Uncompressed, Noisy, Clear, Artifacts Present, Low Bitrate, High Bitrate.

- **Motion and Blur Conditions:** Motion Blur, Static, Camera Shake, Panning Blur, Zoom Blur, Rotational Blur, Linear Motion Blur, Radial Blur, Gaussian Blur, Lens Blur, Out of Focus, Directional Blur, Velocity Blur, Partial Motion Blur, Dynamic Motion, Slow Shutter Speed, Fast Shutter Speed, Artificial Blur, Natural Motion Blur, Vibration Blur.

- **Cultural Differences:** Traditional Clothing, Cultural Festivals, Religious Practices, Food and Cuisine, Architectural Styles, Language and Script, Art and Crafts, Music and Dance, Rituals and Ceremonies, Holidays and Celebrations, Sports and Games, Historical Sites, Marketplaces, Transportation Methods, Housing and Living Spaces, Social Gatherings, Cultural Symbols, Handicrafts, Traditional Instruments, Educational Systems, Work Practices, Family Structures, Social Norms and Etiquette, Festive Decorations, Local Customs.

- **Noise Conditions:** Gaussian Noise, Salt and Pepper Noise, Poisson Noise, Speckle Noise, Impulse Noise, Uniform Noise, Multiplicative Noise, Additive Noise, Quantization Noise, Periodic Noise, Thermal Noise, Shot Noise, Film Grain Noise, ISO Noise, Color Noise, Chromatic Aberration, Background Noise, Low-Frequency Noise, High-Frequency Noise, Random Noise.

- **Occlusion Conditions:** Partial Occlusion, Full Occlusion, Foreground Occlusion, Background Occlusion, Natural Occlusion (e.g., trees, leaves), Artificial Occlusion (e.g., buildings, vehicles), Human Occlusion (e.g., hands, body parts), Animal Occlusion, Object Occlusion, Self-Occlusion (object blocking parts of itself), Motion Occlusion, Temporary Occlusion, Permanent Occlusion, Shadow Occlusion, Transparency Occlusion (e.g., through glass), Blurred Occlusion, Static Occlusion, Dynamic Occlusion, Edge Occlusion, Overlapping Occlusion.

## A.2 DETAILED DATASET AND EVALUATION METRIC DESCRIPTION

The CIFAR-100 dataset serves as a foundational resource for evaluating TIL techniques. It encompasses 100 classes with 600 images each, offering a diverse and substantial platform for performance assessment across various tasks. To facilitate fair comparisons with existing TIL methods, we utilize the ImageNet-Rendition (ImageNet-R) dataset Russakovsky et al. (2015); Wang et al. (2022b). ImageNet-R consists of a wide array of modified images from the ImageNet dataset, enabling comprehensive evaluations of model adaptability and robustness in different visual contexts. Additionally, we employ the TinyImageNet dataset Le & Yang (2015), which is divided into 100 distinct tasks. This dataset allows for a thorough examination of our model's adaptability within constrained environments, featuring 1,000 training samples and 100 testing samples per class. Furthermore, ImageNet100 Deng et al. (2009), comprising 100 classes distributed across 50 tasks, provides an additional framework to evaluate TIL performance. This dataset is particularly valuable in scenarios involving a larger number of tasks, thus facilitating a detailed analysis of model behavior in sequential learning challenges. In summary, the CIFAR-100, ImageNet-R, TinyImageNet, and ImageNet100 datasets collectively provide a comprehensive suite for evaluating the adaptability and robustness of TIL methods across diverse and challenging visual contexts.

In the context of TIL, we employ two prevalent metrics to gauge performance: Average Accuracy and Forgetting. Higher values of Average Accuracy indicate superior performance, while lower values of Forgetting denote better retention of previously learned knowledge Lopez-Paz & Ranzato (2017). In our experiments, we adhere to the default configurations established in previous work, specifically following the task splits and experimental settings outlined in methods such as Dai et al. (2024). This ensures that our evaluations are grounded in established benchmarks, allowing for meaningful comparisons with existing literature. Furthermore, our approach to test-time model adaptation aligns with the methodologies proposed in recent works, particularly those exemplified by Cho et al. (2023). By maintaining consistency with these settings, we facilitate an accurate assessment of our model's performance under varying conditions, which is crucial for understanding its adaptability and resilience in practical scenarios. To summarize, the use of Average Accuracy and Forgetting metrics, alongside adherence to established task splits and experimental configurations, enables a robust evaluation of our model. This approach ensures that our results are comparable to existing studies, thereby providing a meaningful context for assessing the performance and robustness of our proposed methods in TIL.

## A.3 THE T-SNE VISUALIZATION OF THE IMAGENET100 TEST SAMPLES

The t-SNE visualization of the ImageNet100 test samples in the CK space demonstrates clear separation of the 100 classes. The embedding semantic features, projected into a 2D space, distinctly separate each class within the tasks, with different markers denoting different classes. The excellent

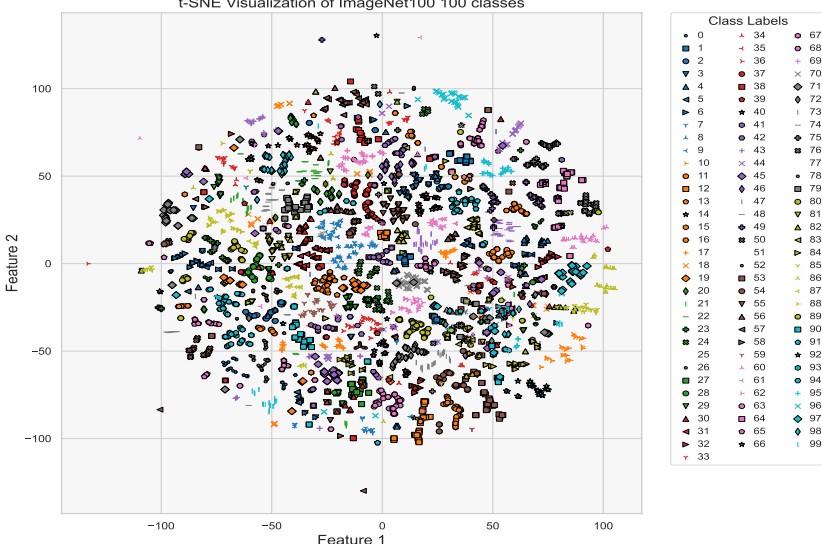

Figure 4: The t-SNE visualization of the ImageNet100 test samples in the CK space demonstrates clear separation of the 100 classes. The embedding semantic features, projected into a 2D space, distinctly separate each class within the tasks, with different markers denoting different classes. The excellent within task prediction performance in the semantic space helps the model achieve superior TIL accuracy.

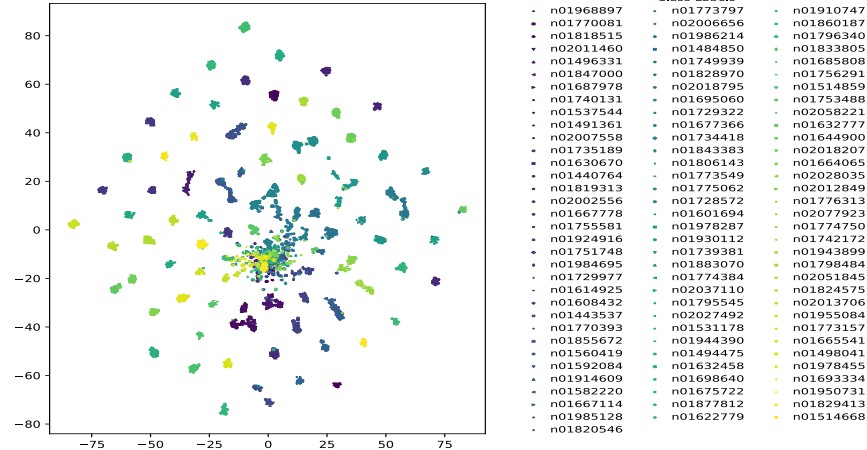

Figure 5: $t$-distributed Stochastic Neighbor Embedding ($t$-SNE) visualizations of features from the proposed framework, as applied to the test set of ImageNet100, reveal the framework's effectiveness in distinguishing among 100 classes spread over 50 independent tasks, with each class represented by 100 samples. In these visualizations, dots of various colors and shapes represent the 100 unique classes within ImageNet100. Notably, despite the significant number of tasks and classes, these elements are predominantly well-separated within a unified feature space. This clear demarcation highlights the framework's ability to achieve remarkable task and class separation, effectively addressing the challenges of class incremental learning. The visual evidence thus supports our method's competency in navigating the complexities of task prediction and underscores its robustness in managing the intricacies of class incremental learning scenarios.

within task prediction performance in the semantic space helps the model achieve superior TIL accuracy.

## A.4 Symbols and Their Meanings

| Symbol | Meaning |
|---|---|
| $l$ | Sample index in the text modality |
| $P$ | Probability Density Function (PDF) value |
| $\mathbf{K}$ | Kernel function |
| $\mathbf{K}_s$ | Kernel based PDF value for an image sample |
| $\mathbf{K}_{text}$ | Kernel based PDF value for the text modal |
| $i$ | Class label index |
| $t$ | Task label |
| $y$ | Class label |
| $j$ | Context index |
| $N_t$ | Total count of the samples in task $t$ |
| $N_i$ | Total number of instances in class $i$ |
| $d$ | Semantic embedding dimension |
| $N_j$ | Total number of instances in context $j$ |
| $h$ | Bandwidth in the kernel density estimation |
| $\mathbf{x}_s$ | Feature embedding in the image modality |
| $\mathbf{x}_{text}$ | Feature embedding in the text modality |
| $s$ | Sample index in the image modality |
| $\mathbf{R}$ | Semi-definite matrix for semantic feature metric learning |
| $\mathbf{L}$ | Linear projection function |
| $\boldsymbol{\mu}_i$ | Mean vector for class $i$ in the text modality |
| $\boldsymbol{\sigma}_i$ | Variance vector for class $i$ in the text modality |

Table 5: Symbols and Their Corresponding Meanings

## A.5 Hyper-parameter Settings

The dimensionality of the semantic feature space in the CK transformation is a crucial parameter that significantly impacts both model performance and computational complexity. To determine the optimal dimensionality, we conducted experiments using the *ImageNet100* dataset. Our findings indicate that low-dimensional representations result in substantial information loss, impeding the model's ability to capture essential variations inherent in the raw features generated by VLMs. Conversely, excessively high-dimensional feature spaces can degrade training efficiency and numerical stability, making the model prone to overfitting and instability. We identified that a dimensionality of 128 provides an optimal balance, offering sufficient capacity to represent the data while mitigating these risks. Beyond dimensionality, we also examined the impact of the bandwidth parameter within the CK framework, testing values ranging from 0.1 to 6.0. Our experiments revealed that a bandwidth setting of 1.0 delivers optimal performance on the *ImageNet100* dataset. Thus, we adopted this value as the default setting, ensuring the model maintains a robust representation of underlying data distributions. Furthermore, we investigated the CK margin parameter, denoted as $\delta$. We found that setting the margin to $(|P_i| + |P_j|)/2$, where $|P_i|$ and $|P_j|$ represent the ranges of the CK in logarithm format for classes $i$ and $j$, respectively, provides a suitable default. This approach ensures a balanced margin that adapts to varying class distributions, enhancing the model's generalization capabilities across different domains.

## A.6 The training process and the convergence analysis

Our proposed learning loss bears some resemblance to traditional hinge loss; however, it significantly diverges in its formulation and intent. The primary objective of our loss function is to align feature representations with their corresponding text modal counterparts. Specifically, the positive samples consist of the feature distributions from the image modality that belong to a given class, while the anchors represent the same class within the text modality. Conversely, the negative samples are drawn from images of different classes.

Throughout the training process, the dynamics of the loss function evolve. The separability of classes varies across tasks, influencing convergence rates. For relatively straightforward tasks, such as

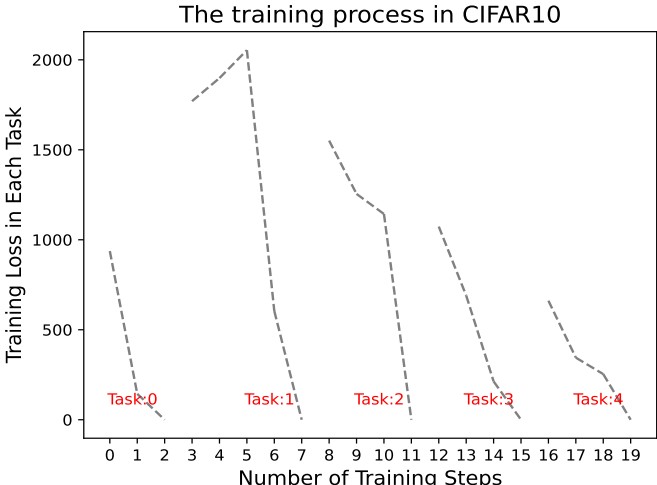

Figure 6: The loss dynamics evolve throughout the training process, with class separability varying by task. For easier tasks, such as classifying dogs and cats, convergence is achieved in fewer epochs due to the effective backbone feature representations. In contrast, for more challenging tasks, like distinguishing between buses and vans, the overlapping feature distributions can lead to confusion, requiring additional training steps for convergence.

distinguishing between dogs and cats, the model typically achieves convergence in fewer epochs due to the robust feature representations provided by the backbone architecture. In contrast, for more complex tasks, such as differentiating between buses and vans, the overlapping feature distributions can lead to increased confusion. Consequently, these tasks require more training steps to reach convergence.

Our approach effectively adapts to the complexities of various classification tasks, ensuring an efficient and effective learning process. Convergence is assured when the margin between the positive and negative distributions exceeds a defined threshold. In extreme cases, some classes are distinctly separated from the outset using the pure backbone features, eliminating the need to train the projection head. Our primary focus is on fine-tuning the projection head to enhance the compactness of distributions within the same class relative to their text modal anchor while pushing the negative distributions further away. This strategy not only reinforces class separability but also promotes robust learning, ultimately leading to improved classification performance within the task and between different tasks.

### A.7  THE FULL ALGORITHM

This approach leverages the flexibility of KDE to adapt to new data distributions dynamically, facilitating effective learning and prediction across numerous task splits. By focusing on clustering within high PDF regions and maintaining separation between tasks and classes, the method aims to optimize performance in a continuous learning scenario, enabling the model to handle new tasks efficiently without forgetting previous knowledge, and the full process is depicted in Alg.1.

```
1  # construct the text modal anchor distribution for the current task
2  for i in range(class_num_in_current_task):
3      # get the text labels for class i
4      index = text_features_labels == i
5      # get the text feature for class i
6      text_samples = text_classes_features[index]
7      # the text embedding for text not require training
8      text_samples = samples.requires_grad_ = False
9      # get the kernalized density estimation for class i
10     k_text = GaussianKDE(X=samples, bw=0.1)
11     # store the k_text to use in training stage
```

```
12      classes_kdes_text.append(k_text)
13
14  # only train the projection head and fix the backbone
15  W_optimizer = optim.SGD(self.W_f.parameters(), lr=1e-6, momentum=args.
        momentum, weight_decay=args.weight_decay)
16  for e in range(1, args.num_epochs):
17      accumulation_steps = 5
18      # for the image samples in the current task, training the projection
19      # head to pull the images near to their text anchors in the same
            class
20      # and push the instance of other class at a margin
21      for it, ((x, label), domain) in enumerate(self.train_loader):
22          x = x.to(device=self.device)
23          label = label.to(device=self.device)
24          # x is the image feature embedding
25          x = self.model_finetuned.encode_image(x)
26          # x_s is the projected feature aiming to match image and text
                modal
27          x_s = self.W_f(x_s)
28          loss = 0
29          # for each text based anchor text distribution
30          for i in range(class_num_in_current_task):
31              # define positive and negative pairs to compare distributions
32              pos_pair_dist = 0
33              neg_pair_dist = 0
34              for j in range(i):
35                  # get projected image features for class j
36                  index = label == j
37                  features = x_s[index]
38                  # measure the distribution distance for the class j image
                        samples and the class i anchor text distribution
39                  dist = classes_kdes_text[i].log_prob(features)
40                  # for the same class, make images close to text
                        corresponding anchor distribution
41                  if i == j:
42                      pos_pair_dist -= dist
43                  else:
44                      # for different classes, image distribution is far from
                            their corresponding text anchor distribution
45                      neg_pair_dist += dist
46                  loss_anchors += torch.relu(pos_pair_dist + neg_pair_dist
                        + margin)
47              loss = loss + loss_anchors
48          # backpropagation for the current batch
49          loss.backward()
50          W_optimizer.step()
51          W_optimizer.zero_grad()
```

## A.8    TRAINING TIME EFFICIENCY

In terms of training time efficiency, our approach demonstrates significant advantages as illustrated in
Fig. 7. We conducted a comparative analysis using the same ViT backbone across ten contemporary
methods: DER++, GSS, ER, GDumb, ASER, SCR, CoPE, DVC, OCM, and OnPro. Notably, our
method efficiently completes five tasks, with each task undergoing 10 epochs, and achieves this in
approximately one minute—each epoch taking merely 1.5 seconds on CIFAR-10 using a single GTX
4090 GPU. This heightened efficiency stems primarily from our method's focus on training only the
linear mapping head. Unlike the other methods that require extensive training across the full feature
space, our approach is designed to address class confusion effectively by only adjusting the linear
mapping to create a mild margin in the PDF space. This targeted training allows for rapid adaptation
without necessitating modifications to the underlying feature space, thereby substantially reducing
the overall computational load and training time. This strategic focus not only enhances efficiency

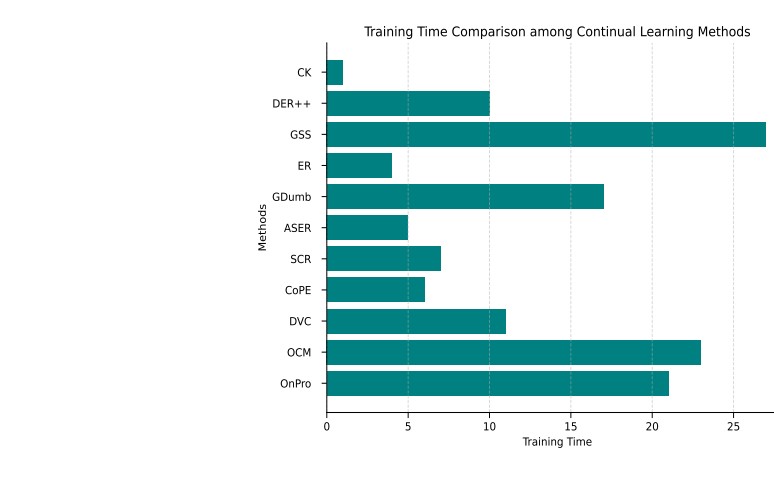

Figure 7: Training Time Efficiency Comparison (In Minutes). Our method completes 5 tasks with 10 epochs per task in approximately 1 minute and each epoch runs in just 1.5 seconds on CIFAR-10. The training time is much shorter than other methods.

but also preserves the integrity of the feature space, providing a streamlined yet powerful solution for task and class prediction in CL settings.

To measure the time to get context feature embeddings, we first utilize the LMM to encode the context for each image. Our timing evaluations indicate that generating the context embedding for a single image within each group takes approximately $0.0145$ seconds. Given that we have 11 context groups and sample 1000 images from the training set for each task, the total time required for context representation is calculated as follows:

$$1000 \, \text{images} \times 11 \, \text{context groups} \times 0.0145 \, \text{s/image} = 159.5 \, \text{s}$$

This duration aligns well with the median processing times observed across the methods evaluated.

It is important to note that as we increase the sample size beyond $1000$ images per task, we anticipate additional overhead for each task due to the increased computational load. Nevertheless, our experiments demonstrate that a sample size of $1000$ images during the training phase is sufficient to achieve satisfactory performance. This finding underscores the efficiency of our approach, balancing computational demands with the effectiveness of the context representation.

The overall time including the training and the context embedding for each task is 3.6 minutes. The comparison with other methods with context embedding (CE) is listed in Fig. 8.

## A.9 BACKGROUND

**Task Incremental Learning (TIL)** has become a crucial research focus, aiming to create models that can learn sequential tasks while mitigating the risk of catastrophic forgetting. TIL methodologies are typically classified into three primary categories: regularization-based methods Aljundi et al. (2018), rehearsal-based methods Chaudhry et al. (2019b), and architecture-based methods Loo et al. (2020). An innovative and more parameter-efficient avenue in TIL is the use of Prompt-based methods Wang et al. (2021). These methods utilize VLMs (VLMs) to learn prompts that direct the model for individual tasks. The prompts, which consist of a small number of learnable tokens, enable efficient parameter utilization. Our work advances TIL by proposing an end-to-end learning framework grounded in CK representation learning. This framework optimally represents tasks and classes through CK space representation optimization, with CKs serving as unique fingerprints for each task and class. This enhances task separation and overall performance in TIL contexts. Additionally, our approach effectively learns context-specific representations, filtering out irrelevant

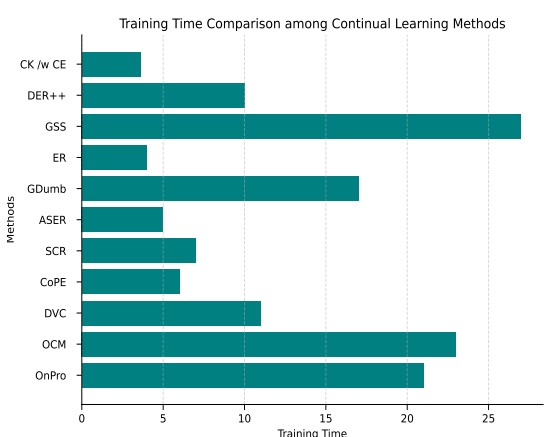

Figure 8: The overall time including the training and context embedding (CE) for each task. The time is still superior to other methods.

contexts to improve class separation within each task, thereby addressing limitations found in both traditional and prompt-based continual learning methods.

**Kernel Density Function Based Representation Learning (KDF-RL)** enhances traditional methods by projecting data into high-dimensional semantic spaces using kernel functions, effectively capturing both linear and non-linear relationships. Key to this domain is Kernel Density Metric Learning, which employs kernel density estimation to establish a probability-based distance metric. The main advantage of kernel-based methods is their ability to model complex, non-linear relationships, with Gaussian kernels effectively representing underlying data distributions to improve classification performance. KDF-RL is particularly beneficial in capturing underlying probability densities, making it advantageous for tasks like anomaly detection where local data density, often assessed via Gaussian kernels, indicates potential anomalies He et al. (2015); Zhang et al. (2018). Furthermore, KDF-RL is effective in metric learning with probabilistic labels, as kernel density estimates help manage label uncertainty, leading to robust distance metrics Huai et al. (2018). Additionally, KDF-RL addresses uncertainty and class-specific variances using methods like Non-isotropic von Mises-Fisher (nivMF) distributions, which model class proxies to capture complex variances and enhance generalization performance Kirchhof et al. (2022). This ability to manage uncertainty and variances is crucial for improving model robustness and adaptability across various learning scenarios.

**Semantic Guidance in the Fine-tuning of VLMs** has garnered significant attention, particularly in open set learning, zero-shot learning, and metric learning. In open set learning, models like CLIP Radford et al. (2021b) develop a vision encoder that aligns with language embeddings, enabling generalization to new classes without labeled visual data Radford et al. (2021a); Ghiasi et al. (2022). Zero-shot learning further leverages this alignment by employing word embeddings from VLMs and knowledge graphs to capture semantic similarities, allowing for inference of unseen classes by measuring distances between vision and language features Naeem et al. (2022; 2023; 2021); Khan et al. (2023). The incorporation of language supervision into vision models facilitates efficient adaptation to new classes within a shared semantic space. Building on these advancements, our work employs kernel-based techniques to enhance representation learning, specifically for the CK task. This approach effectively captures complex relationships and manages uncertainties related to probabilistic labels. By harnessing the strengths of kernel methods, we significantly improve performance in representation learning and related tasks, providing a robust framework for adapting to new semantic classes and enhancing overall model efficacy.

## A.10 KERNEL DENSITY-BASED REPRESENTATION LEARNING

In the kernel density metric learning, the distance in the semantic space can be elegantly expressed using the kernel function $\mathbf{K}$. We define $\mathbf{R} := \mathbf{L}\mathbf{L}^{\top} \in \mathbb{R}^{d \times d}$. Here, $\mathbf{L}$ serves as the projection matrix

that transforms the original feature space of VLMs to the semantic space, facilitating comparisons across tasks within the same space. In the semantic space, the probability metric for class $i$ is formulated as follows:

$$P_i := \frac{\exp(-\|\mathbf{K}_{s,i} - \mathbf{K}_{text,i}\|_{\mathbf{R}}^2)}{\sum_{i \neq i'} \exp(-\|\mathbf{K}_{s,i} - \mathbf{K}_{text,i'}\|_{\mathbf{R}}^2)}. \tag{8}$$

where $\mathbf{K}_s, \mathbf{K}_{text} \in \mathbb{R}^d$ represent the kernelized semantic vectors of the image modal samples and the text modal training instances, respectively. The objective is to bring the training samples in image-modality closer to the text modal distributions when the class labels are identical (i.e., for the same class $i$), while ensuring that different categories (such as $i$ and $i'$) are pushed further apart. To optimize the metric $\mathbf{R}$, we design a projection network that is appended to the VLMs, allowing for the computation of the gradient of the objective function with respect to $\mathbf{R}$.

To find the optimal $\mathbf{R}$, we employ the projected gradient descent method. This methodology facilitates the adaptation of the distance metric in the kernel-induced feature space, enhancing class separation while accounting for the intricate relationships captured by the kernel function. In our implementation, we design a linear projection network to represent $\mathbf{L}$ and learn the projection accordingly.

## A.11 LIMITATIONS

While our semantic-based projection methods and Contextual Kernels (CKs) demonstrate significant advancements in TIL, several limitations remain. First, the reliance on high-quality semantic representations assumes the availability of extensive and well-labeled datasets, which may not always be practical. Second, although our approach mitigates task confusion and semantic collapse, the computational overhead during fine-tuning and the requirement for generating detailed contextual descriptions can be resource-intensive. Third, the method's effectiveness in real-world applications, particularly in highly dynamic environments, warrants further exploration. Lastly, while confidence scores help in abstaining from out-of-category classifications, the mechanism for determining these scores can be refined for greater accuracy and reliability in critical applications.

---

**Algorithm 1** Training Framework Using CK for TIL

---

**Require:** Dataset $\mathfrak{D}$, Vision Transformer Backbone, initial bandwidth $h$
**Ensure:** Trained model with optimized projection head for each task and each class within every
    tasks
 1: Initialize Vision Transformer backbone with pretrained weights
 2: Initialize projection head parameters randomly
 3: **for** each task $t$ **do**
 4:    Evaluate the context of the current task and get the text-modal anchor class distribution $K_{text}$
      based on kernel density estimation (without test samples)
 5:    Extract raw features $\boldsymbol{X}_s$ using frozen LMM models for the image samples
 6:    Project features $\boldsymbol{X}_s$ into $d$-dimensional space
 7:    **for** each class $i = 1$ to $m$ **do**
 8:      Measure the distribution distance for the image samples to each anchor class $\boldsymbol{K}_{text}$, and
        train the projection network to draw the images close to the anchor distribution in text modal
        and push samples away from other class anchors.

$$\mathcal{L}(\mathbf{L}) = \max(- \sum_{x_{text} \in D_i} \mathbb{1}_{\{y=i\}} \mathbf{K}(\mathbf{x}_s - \mathbf{x}_{text})$$
$$+ \sum_{x_{text} \in D, x_{text} \notin D_i} \mathbb{1}_{\{y \neq i\}} (\mathbf{K}(\mathbf{x}_s - \mathbf{x}_{text}) + \Delta, 0) \tag{9}$$

 9:    **end for**
10:    Perform back-propagation and update the projection head parameters
11: **end for**
12: Evaluate model on validation set to adjust $h$ if necessary
13: **for** each test sample $\mathbf{x}$ **do**
14:    Compute the context based task representation and class distribution in image modal.
15:    Classify the test sample $\mathbf{x}_s$ to each task representation and selecting the correct task.

$$\mathcal{T} = \arg\max_t \sum_{i \in \mathcal{Y}^t} \arg\max_i \mathbf{K}_i(\mathbf{x}_s), \tag{10}$$

16:    Classify the correct class id under the current task.

$$P[Y = i | \mathbf{x}_s, \mathcal{T}] = \frac{\mathbf{K}_i(\mathbf{x}_s)}{\sum_{i' \in \mathcal{Y}^{\mathcal{T}}} \mathbf{K}_{i'}(\mathbf{x}_s)}, \tag{11}$$

17: **end for**

---

