# OpenReview forum: "Contextual Kernels for Task-Aware Fine-Tuning in Vision-Language Models"
_ICLR.cc/2025/Conference — Submitted to ICLR 2025_

### Official Review · Reviewer_k3oA · 2024-10-27

**Soundness:** 3
**Presentation:** 3
**Contribution:** 3
**Rating:** 6
**Confidence:** 3

**Summary:**

This paper introduces a Contextual Kernel Density-Based Task Representation Learning Framework to enhance the performance of VLMs for Task Incremental Learning. This method constructs Contextual Kernels (CKs) in the semantic space, enabling task-specific class separation and enhanced model adaptability during fine-tuning. Authors conducted extensive experiments on multiple datasets and the results validate the effectiveness of this approach.

**Strengths:**

- Originality: The proposed CK framework provides a new approach to TIL by focusing on contextual kernels for task-specific fine-tuning.

- The overall presentation is clear.

- The performance of the CK framework is superior to that of existing methods.

**Weaknesses:**

- The proposed CK framework needs kernel computing, which might be time-consuming. Could authors provide a training time comparison of CK and other methods to demonstrate the training efficiency of the CK framework?

- The implementation is unclear. Since the authors did not provide any code, could authors further provide a pseudo-code to better understand how authors implement their framework?

- The form of $P$ in Eq. 8 is different from other parts.

**Questions:**

see weaknesses

---

> ### Author Response · Authors · 2024-11-18
> **Address the concerns**
>
> Thank you for your insightful and encouraging comments on our paper. We truly appreciate your feedback, which has helped us enhance the clarity and quality of our work. Below, we address your specific concerns:
> 1) Kernel Computing in the CK Framework: We acknowledge that the CK framework relies on kernel computing. To clarify, the samples used to estimate the distribution for each class do not exceed 3,000, as we sample accordingly. Our approach constructs kernel-based estimations for each class utilizing both text and image modalities, ensuring that the process remains efficient and fast. Moreover, during the training phase, only the projection head is updated, resulting in a significantly shorter training time. We would like to emphasize that the overall computation time, when compared to other methods, is not substantially longer. We will include a detailed time comparison in the revised manuscript to support this point.
> 2) Figure Expression and Algorithm Description: We appreciate your feedback regarding the expression of our work's streamline in the figures. In response, we have included a comprehensive algorithm description and relevant code samples in Appendix 8. We believe that these additions will offer a clearer and more thorough understanding of our approach, thereby addressing the shortcomings you identified.
> 3) Equation Consistency: We have revised the form of the equation in Eq. 8 to maintain consistency throughout our paper, as you suggested. Your attention to detail in this matter is greatly appreciated, and we are committed to ensuring that our presentation is clear and coherent.
> Once again, we sincerely thank you for your valuable insights. Your contributions have been instrumental in improving our manuscript, and we hope that our revisions meet your expectations. We look forward to your continued feedback.

---

> > ### Comment · Reviewer_k3oA · 2024-11-18
> > **Thank you for detailed rebuttal.**
> >
> > Thank you to the authors for their detailed responses. My concerns have been addressed to some extent. I will maintain my positive score for now and may revise it after the authors provide updates on the training efficiency comparison and after considering the responses from other reviewers.

---

> ### Author Response · Authors · 2024-11-18
> **About the time efficiency**
>
> Thank you for your timely response and for your thoughtful comments on our work. We truly appreciate your insights.
>
> We would like to clarify the concerns regarding the overhead of our model. As detailed in Appendix 9, the time efficiency metrics indicate that the context embedding time for processing 1000 sampled training or test images on the CIFAR-10 dataset for each task is approximately 2.6 minites, while the training time is notably efficient at just 1 minutes.
>
> Furthermore, we have conducted a comparison with other methods under the same experimental settings, which reveals that our approach maintains competitive time efficiency.
>
> We hope this information addresses your concerns and highlights the efficiency of our approach. Thank you once again for your valuable feedback.

---

### Official Review · Reviewer_whEd · 2024-11-03

**Soundness:** 3
**Presentation:** 2
**Contribution:** 3
**Rating:** 6
**Confidence:** 3

**Summary:**

This paper introduces the concept of context kernels (CKs) to deal with Task Incremental Learning scenarios. Specifically, the paper proposes a framework that operates in two stages: 1) Generate CK representation for each class using a context prompt pool; 2) Train a projection network to learn the semantic representation and effectively separate different tasks and classes in the semantic kernel space. A threshold CK-based confidence check is further introduced to filter out task-irrelevant test samples during inference and ensure robustness in safety-critical applications. Extensive experiments over a wide range of datasets show that the proposed approach achieves significantly better performance over existing techniques.

**Strengths:**

1. The paper provides a comparison of the proposed method against a number of competitive prior works, and carried out experiments on various datasets for the TIL task, which is sufficient to validate the approach. The empirical results demonstrate that the framework achieves significantly better performance than previous techniques.

2. The CK-based confidence score is a clever way to make models refrain from making decisions over input samples outside the scope of the tasks that it learns to handle. This module allows for the possibility of application in safety-critical scenarios such as autonomous driving and medical diagnostics, adding to the practicality of the paper.

3. The paper offers an interesting observation on the semantic distribution of different datasets in the introduction, efficiently displaying the motivation behind the framework design.

**Weaknesses:**

1. The formulation of the proposed framework as well as presentation of each module is convoluted at times and not easy to follow. In my view, the paper needs to put more effort into explaining the entire pipeline without jumping into the specifics first. Additionally, too much space is dedicated to describing the framework design and the analysis on experimental results is too brief. In Section 4.1, the paper says “…Tables 1 illustrate that our proposed method consistently outperforms existing techniques…”. Perhaps it should be “Tables 1 and 2” since Table 1 only shows the results on ImageNet-R. Another crucial section missing is ablation study on different modules, which seems not to be included anywhere in the paper.

2. The figures in the paper are aesthetically unappealing and doesn’t really highlight the streamline of the framework to help readers understand. Especially in Figure 1, the figure is not centered and some of the fonts even overlap with the title.

3. Throughout the paper, “generality” is consistently used, but the correct word should be “generalizability” or “generalization capability”.

**Questions:**

Do we need to hand-craft the word list in the context prompt pool, or is that done somehow automatically? The paper claims that the context prompt pool is extensible. It would be nice if the authors could shed some light on how to do so.

---

> ### Author Response · Authors · 2024-11-18
> **Addressing our concerns**
>
> Thank you for your insightful feedback on our manuscript. We appreciate your suggestions and have made several adjustments to better clarify our contributions. Below are our responses to your concerns:
> 1) Framework Design Description: We acknowledge that the framework design description in Figure 2 may not fully convey the necessary details due to space limitations. In response to your suggestion, we have clarified the focus of our paper and prioritized a more coherent introduction to our concepts before delving into specifics. To address this shortcoming, we have included a comprehensive algorithm description and code samples in Appendix 8, which we believe will provide a clearer overall understanding of our approach.
> 2) Experimental Results: We have moved Table 2 to Appendix 2 for better organization. Additionally, we have included more experimental results in the appendix, such as visualizations of task and class prediction outcomes, along with detailed discussions to enhance the interpretability of our findings.
> 3) Terminology Update: We have replaced the previous term with the correct word “generalizability,” as per your suggestion. Thank you for pointing this out.
> 4) Context Expansion: Appendix 1 now includes a comprehensive context list with 11 categories. We also plan to expand the context using the API of the LLM, providing example test images during the testing phase with the prompt: ``Provide more specific context that is unique to the images of the current task but not directly related to the classes.'' This will help to enhance the contextual richness of our analysis.
>  5) Ablation Study: We have included the ablation study in Appendix 6. This addition aims to elucidate the impact of various components of our framework, further validating our methodology.
> 6) Figure Clarifications: The main streamline graph is presented in Figure 2, while Figure 1 is dedicated to illustrating our motivation. We emphasize that the context of the task plays a crucial role in the model’s adaptability, and we have made this connection clearer in our revised manuscript.
> We believe that these revisions effectively address your concerns and enhance the overall clarity and contribution of our work. Thank you once again for your valuable feedback, and we look forward to your further thoughts.

---

> > ### Comment · Reviewer_whEd · 2024-11-18
> > **Thank you for your detailed rebuttal.**
> >
> > Thank you to the authors for your timely and detailed response. I appreciate the clarification on the core algorithm/framework design, as well as the additional experimental results, which helped strengthen the claim of the paper. Most of my concerns have been addressed, however, I still hope that the authors could revise the figures in the paper so they contain more details and are more asthetically pleasing to the readers. Moreover, most of the paper's analysis lies in the appendix instead of the main body of work, making the overall arrangement seems rather unbalanced (too focused on the methodology and too little for the rest). I'm inclined to change my score into a positive one, but will maintain it for now and wait for more updates from the authors & responses from other reviewers.

---

> > > ### Author Response · Authors · 2024-11-19
> > > **About reorganization of our paper**
> > >
> > > Thank you for your valuable feedback and suggestions for enhancing the structure of our paper. We appreciate your insight regarding the balance between methodology and analysis. In response, we plan to relocate the detailed method descriptions to the Appendix, allowing us to feature key experiments and discussions more prominently in the main body of the paper.
> > >
> > > Additionally, we acknowledge your comments on the figures. We are actively working on redesigning them to improve their clarity and overall aesthetic appeal, ensuring they effectively communicate the information to our readers.
> > >
> > > Thank you once again for your constructive input. We believe these changes will significantly enhance the quality of our manuscript.

---

> > > > ### Comment · Reviewer_whEd · 2024-12-03
> > > >
> > > > The latest version has addressed most of my concerns and the paper's quality has significantly improved. Thus, my final decision is to increase the overall rating to a positive one.

---

### Official Review · Reviewer_XPGc · 2024-11-04

**Soundness:** 3
**Presentation:** 2
**Contribution:** 3
**Rating:** 6
**Confidence:** 3

**Summary:**

Generally, VLMs can be used in a zero-shot setting. However, this paper assumes that there are some training samples at test time. More specifically, the paper focuses on the Task Incremental Learning problem (TIL). The authors propose a novel method for test-time fine-tuning of VLMs. Given a test image, a (contextualized) image description is generated. Here context refers to things like view point (e.g. side view, front view, etc.) , style (e.g. art, sketch, etc.) or background (e.g. sky, forest, beach, etc). The key challenge is to balance between maintaining generality (not forgetting old tasks) and adapting to the new tasks. To maintain the generality, the VLM is frozen. To adapt to new tasks a projection matrix (a linear layer) is learned at test time through gradient decent. The loss function is a hinge loss rather than the usual cross-entropy loss.

**Strengths:**

The method is fairly simple.
Comparison is done with a good number of the previous methods (10 methods), although I have some concerns about datasets (more on this will follow).

**Weaknesses:**

It seems that a good chunk of the experimental results on several data sets is completely missing. Table 1 shows the performance on the imagenet-R data set. I can’t seem to find similar tables for Cifar-100 and tiny-imagenet and imagenet-100 datasets.

Use of a predefined threshold on K(xs) to identify outliers and exclude them from training is not well justified. My concern is that at the beginning of the training process, where the projection matrix L is randomly initialized,  K(xs) values are poor estimates of the true similarity between an image and text and early exclusion of some training samples by thresholding based on such a noisy estimate of similarity may not be reliable.

All the experiments are done using ViT B/16 backbone. Performance using backbones other than ViT B/16 will provide more insight into the robustness of the proposed method.

On page 6, N_j is defined as the number of pre-defined context. It is not clear what that means and if it is predefined. More broadly, the authors use the term “context” throughout the paper without clearly explain what that mean. A clear example would help. Or at least properly explain Figure 2. The brief list provided in the appendix feels disconnected from the main body of the paper.

In eq4, Ks is for text modality and Kt is for image. But in explanation that follows eq 5, s is for image modality and t is for text. In the table of symbols in A4, l is for text, t is for task and s is for image. Please clarify the notation.

**Questions:**

The loss function is a margin base or hinge loss (Eq6) instead of the usual cross-entropy loss. When using a hinge loss usually the choice of negative samples affects the convergence. Did the authors face any issue in regards to convergence and the choice of negative samples in the hinge loss?

---

> ### Author Response · Authors · 2024-11-18
> **Clarification of Contribution and Addressing the Concerns**
>
> 1. **Clarification of Contribution**:
>    There may be a gap between the reviewer's summary and our paper's focus due to our paper's unclear expression. To clarify the contributions of our work, we enhance the paper with several key points:
>    - The primary aim is to design a task and class representation learning framework that predicts task and class labels at test time using an independent projection head for each task. Our method effectively differentiates non-overlapping tasks and classes within the current task by formulating the classes with a context kernelized feature distribution representation.
>    - To further illustrate our approach, we provide an overall depicting graph in **Figure 2** and a detailed algorithm description along with key code in **Appendix A.7**. In the training stage we train an independent projection head and kenelized class distributions for each task, while in the test stage, we use test time context kernel representation to find the correct trained task head and the correct class distribution for prediction.
>    - In contrast to other methods, our approach uniquely does not require any test images during the training phase. Instead, we sample some images to derive context embeddings in the text modality, which are subsequently used to establish a kernelized density distribution that serves as training anchors. During the testing phase, upon receiving the test samples, we again sample some images to obtain context embeddings. These embeddings are utilized to identify the relevant task by analyzing distribution characteristics, which facilitates task and class predictions. While test-time adaptation is a well-established practice in the field, our methodology distinguishes itself from this conventional approach. We emphasize that our process is designed to leverage the contextual information more effectively, thus enhancing predictive performance while maintaining operational efficiency.
>
>    Additionally, our loss function is based on distribution rather than traditional samples: the anchor is the text modal distribution considering related context, the positive is the image modal distribution for the same class as the text modal (not training samples), and the negative is the image distribution for other classes. This differs from traditional hinge loss, which includes both positive and negative samples.
>
> 2. **Performance Analysis**:
>    The performance analysis for the Tiny ImageNet and ImageNet-100 datasets can be found in **Table 3** of our experimental section. For CIFAR-100, additional results are provided in **Table 2**.
>
> 3. **Training Process Explanation**:
>    During training, our projection head optimizes the relationship between the text modal and image modal distributions, rather than samples. Since the backbone already provides sufficient representation ability in the global classification space, our projection head fine-tunes the model to make the distributions for the same class more compact and separate for different classes within the same task. Our focus is to optimize the class distribution representation in the local classification space for the current task with context based text modal anchor distribution. For example, distinguishing between dogs and cats requires fewer epochs, while differentiating between horses and ponies—which are more confused in the global classification space—takes more training steps. This training convergence is illustrated in **Figure 6**.
>
> 4. **Definition of 'Context'**:
>    In our paper, the term "context" refers to information that aids classification but is specifically present in the current task excluding class information, such as lighting conditions, styles, backgrounds, environments, etc. A full list can be found in **Appendix 1**.
>
> 5. **Ablation Study on Backbones**:
>    We conducted experiments using various backbones, with the ablation study presented in **Table 4**. This highlights the significance of the backbone's feature representation ability in our method.
>
> 6. **Notation Clarification**:
>    In our paper, we denote the image features as $ X_s  $ and the distribution of the image as $ K_s $. For the text modal, the features are denoted as $ X_{text} $ and the distributions as $ K_{text} $. The label $ t $ represents the task index, while $ i $ denotes the class index for the current task throughout the paper.
>
> 7. **Loss Function Design**:
>    Our loss design diverges from traditional hinge loss; we represent the anchor, positive, and negative not based on samples but through the text modal distribution and the image modal distribution across the same and other classes within the current task. A detailed convergence analysis is provided in the **Appendix A.6**, along with thorough explanations.
>
> We appreciate your insightfull comments and hope this response clarifies our contributions and findings effectively. Thank you for your consideration!

---

> > ### Comment · Reviewer_XPGc · 2024-12-03
> >
> > I went over the responses and other reviewers comments. I appreciate the authors efforts in clarifying and addressing my comments. I increased my rating.

---

### Meta-Review · Area_Chair_F3Tm · 2024-12-20

**Metareview:**

The paper is of poor quality and has several notable weaknesses that were not identified in the reviews. First of all, the results presented in this work have a big problem. Specifically, Table 1 and Table 2 are supposed to show results on two different datasets, namely ImageNet-R and CIFAR-100, but the numbers in these tables are 100% identical. This is a big mistake and the numbers need to be revised, which means another round of review is needed. Second, the evaluation and comparisons are unfair. The method is clearly based on a VLM (which is CLIP, the AC assumes; because no detail was given about the specific architecture). However, the baselines compared in the tables are based on pure vision models and therefore the comparisons are unfair (CLIP was pre-trained on 400M image-text pairs covering much broader knowledge than other pre-training datasets like ImageNet). Third, the writing is unclear in many places. For instance, Section 2 is supposed to discuss the background, as suggested by the section title. But instead the secion just reiterates the contributions without giving enough context, and no citations are provided for those mentioned works (which is unprofessional). The methodology part also has lots of problems: no detail for the mapping network introduced in fig.2 and no discussion on how this impacts the results; no detail for the kernel function (eq.4); no detail for the threshold value; and so on. Overall, the paper is incomplete and needs to be significantly revised for another round of full review. After discussion with the SAC, the AC decides to overrule the reviewers' decisions and reject this paper.

**Additional Comments On Reviewer Discussion:**

After going through the reviews and the rebuttal, the AC has no idea why the reviewers found the rebuttal "significantly improved" the paper's quality and gave 3x accept.

---

### Decision · Program_Chairs · 2025-01-22

Reject